# Association of Cardiometabolic Multimorbidity Pattern with Dietary Factors among Adults in South Korea

**DOI:** 10.3390/nu12092730

**Published:** 2020-09-07

**Authors:** Dawoon Jeong, Jieun Kim, Hansongyi Lee, Do-Yeon Kim, Hyunjung Lim

**Affiliations:** 1Department of Medical Nutrition, Graduate School of East-West Medical Science, Kyung Hee University, Yongin 17104, Korea; ekdns0907@naver.com; 2Research Institute of Medical Nutrition, Kyung Hee University, Seoul 02447, Korea; jieunkim@khu.ac.kr (J.K.); flowerlhsy@hanmail.net (H.L.); rou_@naver.com (D.-Y.K.)

**Keywords:** cardio-metabolic disease, dietary factors, multimorbidity pattern, South Korea

## Abstract

Globally, cardiometabolic multimorbidity pattern (CMP) is a complex chronic health status that negatively effects the life expectancy of adults globally, even more than single diseases. We aimed to identify multimorbidity patterns in Korean adults to clarify the associations between dietary factors and CMP. Nationally representative data of 9011 Korean adults aged 19–64 years were obtained from the Korean National Health and Nutrition Examination Survey (KNHANES) from the period 2013 to 2015. Multimorbidity patterns for CMP, inflammatory disease, cancer and other disease patterns were identified by exploratory factor analysis. Dietary factors including food and nutrient intake and dietary habits were evaluated. Multivariable-adjusted logistic regression models examined the associations between dietary factors and CMP. More than half of the multimorbidity patterns were CMP (*n* = 4907, 54.5%); CMP subjects were more likely to be older, male, less educated, lower income, laborers, smokers, and high-risk consumers of alcohol than those of non-CMP subjects. A higher intake of calcium (OR = 0.809, 95% CI = 0.691–0.945), potassium (OR = 0.838, 95% CI = 0.704–0.998), and fruits (OR = 0.841, 95% CI = 0.736–0.960) were inversely associated with the prevalence of CMP, while the consumption of irregular meals (OR = 1.164, 95% CI = 1.034–1.312) and skipping breakfast (OR = 1.279, 95% CI = 1.078–1.518) was positively related to a 16% and 28% higher likelihood of CMP, respectively. CMP accounts for more than half of the multimorbidity patterns in the Korean population, and lower intake of calcium, potassium, fruits, and skipping meals have strong associations with CMP.

## 1. Introduction

Cardiometabolic multimorbidity is defined as the co-existence of chronic disease such as type 2 diabetes, coronary heart disease, and stroke in an individual [1]. Since life expectancy and economic development have increased in the last century in Asia [2], the prevalence of multimorbidity has been increasing in Korea, supposedly due to “nutrition transition” [3]. In a general practice setting, the prevalence of multimorbidity increased with age in both men and women [1,4]. Multimorbidity increased personal and social burdens including higher disability [5], higher mortality rates [6], longer hospital stays, and higher hospital readmission rates [7,8]. The epidemiology of multimorbidity is not well understood because most studies have examined single-disease-related outcomes or excluded patients with comorbid diseases [9]. A novel approach has been performed to develop an actual evidence-based guideline that offers clinical support and health-care for those who struggle with multiple chronic conditions [9]. In Korea, there are a few studies describing multimorbidity pattern as well as socioeconomic and quality of life but these are limited to the elderly [10,11]. However, cardiovascular and metabolic diseases [3] are rapidly increasing in the Korean adult population that are nonelderly [12].

Epidemiologic studies have suggested that a lack of adequate physical activity [13], smoking [14], high-risk alcohol consumption [15], and unhealthy dietary intake [16] increase the risk of many chronic diseases. On the other hand, several foods and nutrients including fruits, vegetables, whole grains [16], polyunsaturated fatty acids (PUFAs) [17], calcium, and potassium [18] have preventive effects on multimorbidity, whereas sodium intake and salty foods are related to a higher risk of multimorbidity [18]. Some corroborative evidence shows that certain dietary habits also influence the occurrence of multimorbidity [19]. Thus, a multifactorial approach is required for the association of dietary factors with chronic status of multimorbidity in Korean adult population. Our objective was to identify multimorbidity patterns in Korean adults and to examine the associations between dietary factors and cardiometabolic disease pattern (CMP).

## 2. Materials and Methods

### 2.1. Study Population

We used data from the sixth Korean National Health and Nutrition Examination Survey (KNHANES VI) from 2013 to 2015. The KNHANES is a national cross-sectional survey that was performed by the Korea Centers for Disease Control and Prevention and conducted using a multistage, clustered, stratified, and rolling sampling method. The survey consists of a health interview, health examination, and nutrition survey. The study protocols were approved by the Institutional Review Board (IRB) of the KCDC (2013–07CON-03-4C, 2013-12EXP-03-5C, 2015-01-02-6C). All participants provided written informed consent.

We analyzed the data of 22,948 participants collected by the KNHANES VI. Eligible participants were adults aged 19 to 64 years. The exclusion criteria were as follows: pregnant and lactating women, participants with extreme energy intake (<500 kcal/day or >5000 kcal/day), and participants missing values for major diseases, dietary factors, and other covariates. After applying these exclusion criteria, the total number of participants for this analysis was 9011 adults (3883 men and 5128 women) (Figure 1).

### 2.2. Assessment of Dietary Factors

The mean daily total energy, nutrients and food intake were assessed by a 1-day, 24-h recall method based on the 8th Korean Foods and Nutrients Database of the Rural Development Administration [20]. Food groups were classified into 18 groups (grains, potatoes, sugars, beans, nuts, vegetables, mushrooms, fruits, meats, eggs, fish, seaweeds, dairy products, oils, beverages, seasonings, processed foods, etc.) in the Korean Nutrient Database [21]. Dietary habits were assessed using a questionnaire that evaluated four key dietary factors: meal frequency, breakfast fasting, frequency of dining out, and utilization of nutrition labels that affect chronic disease.

### 2.3. Health-Related Behaviors

Health-related behaviors were categorized by the following: smoking status, alcohol drinking, and physical activity. Subjects were categorized as current smokers, former smokers who quit smoking, or non-current smokers who never smoked. For alcohol consumption, subjects were categorized into high-risk drinking and low-risk drinking. The degree of risk was evaluated by the amount of alcohol consumption per week. Alcohol consumption was calculated by the frequency of drinking for one year that was converted into a weekly rate, and then multiplied by the amount of one drink. In accordance to guidelines of KNHANES, the recommended intake for Korean adults, high-risk drinking was judged as drinking more than seven cups per week for females and more than 14 cups per week for males [22]. Physical activity was defined using the metabolic equivalent task [1] score as per the scoring protocol of the Korean International Physical Activity Questionnaire short form [23]. Physical activity was categorized according to the total MET score as inactive (<600 MET-min/week), active (600 to 3000 MET-min/week), or health-enhancing (>3000 MET-min/week).

### 2.4. Definition of Multimorbidity

Multimorbidity was defined as suffering from two or more of the chronic medical diseases [1]. The medical conditions used to define multimorbidity in this study, 24 chronic diseases including hypertension, dyslipidemia, stroke, cardiovascular disease (e.g., myocardial infarction, angina pectoris), osteoarthritis and osteoporosis, cataract, depression, asthma, pulmonary tuberculosis, hepatitis B, cancer, thyroid disease, anemia, sinusitis, allergic rhinitis, atopic dermatitis, and tympanitis using a combination of methods including a self-reported physician diagnosis and/or current use of a disease treatment medication, or health-examination-based diagnosis: (1) Hypertension was defined as systolic blood pressure ≥140 mm Hg or diastolic blood pressure ≥90 mmHg [24]. (2) Dyslipidemia was defined as total cholesterol ≥200 mg/dL, high-density lipoprotein cholesterol <40 mg/dL, or low-density lipoprotein cholesterol ≥100 mg/dL [25]. (3) Diabetes mellitus was defined as fasting blood glucose ≥126 mg/dL [26]. (4) Obesity was defined as body mass index ≥25 m^2^/kg [27]. (5) Anemia was defined as hemoglobin <12 mg/dL in females or <13 mg/dL in males [28]. Intrinsic cardiometabolic risk factors (total cholesterol, LDL-cholesterol, VLDL-cholesterol, insulin, HDL:LDL ratio, HDL:TAG) were not included.

Other diseases were diagnosed according to self-reported results in the health interview of KNHANES.

### 2.5. Multimorbidity Pattern Analysis

To identify multimorbidity patterns in Korean adults, we analyzed exploratory factor analysis, which is a statistical technique used to identify factors by summarizing the correlation between sets of variables and to understand the underlying structure of the data [29]. We identified multimorbidity patterns by using exploratory factor analysis, the most widely used analytical approach to account for binary morbidity data. A tetrachoric correlation matrix will lead to more valid results to assess the correlation structure between the variables—in this case, to assess the correlation with each disease. Particularly due to the dichotomous nature of the variables, a tetrachoric correlation matrix was generated. To increase the epidemiological significance of the study, a prevalence of diseases of more than 1% was included for tetrachoric correlation matrix generation. Then, using the results of this correlation matrix, a factor analysis was conducted. The number of factors extracted by scree plot was utilized, in which the eigenvalues of the correlation matrix were represented in descending order to produce the inflection point of the curve, eigenvalue 1.0. To facilitate interpretation of the factors, it was rotated using the oblique rotation (oblimin) method. The Kaiser–Meyer–Olkin method was implemented to determine the adequacy of the sample in factor analysis. To determine the most appropriate multimorbidity pattern, we selected variables where factor loading was above 0.25 [30], and called patterns according to common features of diseases among the pattern (Appendix A).

About half of subjects had Cardiometabolic multimorbidity pattern (CMP), which consists of obesity, dyslipidemia, hypertension, diabetes mellitus, osteoarthritis and osteoporosis, depression, stroke, and cardiovascular disease, which are characterized as metabolic diseases, and 35.9% of subjects suffered from isolated CMP (Figure 2). In contrast, ‘inflammatory diseases pattern’ [17], which included allergic rhinitis, sinusitis, atopic dermatitis, asthma, and otitis, which is the main cause of inflammation, included 23.1% of subjects, and ‘cancer and other diseases pattern’ [31], which included cancer, anemia, thyroid disease, hepatitis B, and cataract, included 13.0% of the subjects [3,9,32]. Subjects who had more than two patterns were at 17.9%, and 1.4% had three patterns. A total of 11% of subjects had cardiometabolic disease and inflammatory disease at the same time, and subjects who were not included in any of the three multimorbidity patterns included 31.0% of all subjects (Figure 2).

### 2.6. Statistical Analysis

All statistical analyses were performed using PROC SURVEY in SAS version 9.4 (SAS Institute, Cary, NC, USA). Strata, clusters, and weights were determined to reflect the Korean population estimates to consider the complex sampling design. Student *t*-tests and two-sided chi-square tests were used to compare the continuous variable of the age of the subjects and categorical variables of sociodemographic characteristics, health-related behaviors, and dietary habits between CMP and non-CMP groups, which was the frequent multimorbidity pattern.

To test the hypothesis that subjects with CMP, which is the most frequent multimorbidity pattern, ate differently to non-CMP subjects, we compared the mean intake of nutrients and food between non-CMP and CMP subjects using the PROC SURVEYREG procedure adjusting for age, sex, and energy intake (*p* < 0.05). Finally, we tested the hypothesis that unhealthy dietary factors were associated with CMP. To compare levels of nutrient and food intake, we divided the subjects into tertiles. Logistic regression analysis was conducted to estimate the odd ratios (ORs) at the 95% confidence intervals (CIs) of nutrient and food intake, dietary habits, and health-related behaviors using the PROC SURVEYLOGISTIC procedure. Multivariate analysis was performed after adjustment for age, sex, and energy intake (Model 1), as well as income, education level, smoking status, alcohol drinking, and physical activity (Model 2).

## 3. Results

### 3.1. Sociodemographic Characteristics and Health-Related Behaviors

Overall, we analyzed data from 9011 South Korean adults aged from 19 to 64 years. A total of 58.8% of the CMP group was male (mean age: 45.1 ± 0.2 y) and less educated (≥Middle school: 22.1%) than the non-CMP group (*p* < 0.05) (Table 1). The CMP group lived in rural areas (28.9%), had a lower income (lowest: 26.1%) and were more likely to be a laborer than the non-CMP group (*p* < 0.05). More current-smokers (28.7%) and high-risk alcohol consumers (20.8%) were observed in the CMP group (*p* < 0.05) (Table 1).

### 3.2. Mean Daily Consumption of Food and Nutrients

The difference between the two groups for nutrient and food intake is presented in Table 2. A difference in energy intake and percentage of macronutrients was not identified between the groups. There was a significantly lower carbohydrate and calcium intake in the CMP group when compared to the non-CMP group (*p* < 0.05), whereas other nutrients were not significantly different between the two groups. Fruits (192.07 ± 5.40 g vs. 208.70 ± 5.19 g), sugar and sweetener (11.51 ± 0.34 g vs. 13.08 ± 0.44 g) intake in the CMP group were significantly lower (*p* < 0.05) than those of the non-CMP group. On the other hand, higher consumption of beverages was found in the CMP group (370.13 ± 10.01 g vs. 342.62 ± 8.08, *p* < 0.05).

### 3.3. Associations (ORs and 95% CIs) between Food and Nutrients and CMP

After adjustment for age, sex, and energy intake (Model 1), multivariable analysis showed significantly higher ORs for CMP in nutrient intake for the following factors: lower PUFAs, dietary fiber, calcium, sodium, potassium (*p* < 0.05) (Table 3). After further adjusting for economic status and health-related behaviors (Model2), the association with higher intake of calcium (tertile 2: OR = 0.843, 95% CI = 0.734–0.969; tertile 3: OR = 0.809, 95% CI = 0.691–0.945; *p* for trend= 0.018) and potassium (tertile 2: OR = 0.804, 95% CI = 0.696–0.929; tertile 3: OR = 0.838; 95% CI = 0.704–0.998; *p* for trend = 0.013) in the CMP group were remained significant when compared with those who ate fewer nutrients. Among foods, higher intake of fruits and vegetables was associated with lower CMP in Model 1; however, only fruit intake (tertile 2: OR = 0.818, 95% CI = 0.717–0.933; tertile 3: OR = 0.841, 95% CI = 0.736–0.960, *p* for trend = 0.0001) had a negative association with CMP in Model 2 compared with those who ate fewer fruits (Table 3).

### 3.4. Associations (ORs and 95% CIs) between Health-Related Behaviors Including Dietary Habits and CMP

Regarding the association between health-related behaviors including dietary habits and CMP, subjects who had a two-times-per-day meal frequency (OR = 1.164, 95% CI = 1.034–1.312) had higher CMP compared with those who regularly ate three meals daily (Table 4). In addition, irregular breakfast consumers who ate breakfast once or twice a week (OR = 1.279, 95% CI = 1.078–1.518) were associated with higher CMP compared with regular breakfast consumers that consume breakfast every day. Among health-related behaviors, physical activity was not significantly associated with CMP, but smoking (OR = 1.303, 95% CI = 1.108–1.533) and alcohol drinking (OR = 1.490, 95% CI = 1.292–1.718) were associated with CMP and remained significant after adjustment for all covariates.

## 4. Discussion

We identified multimorbidity patterns by using exploratory factor analysis, the most widely used analytical approach, to account for the correlation structure between the variables—in this case, to assess the correlation with each disease. Based on a nationally representative dataset of South Korean population, over half of Korean adults who, aged 19–64 years, have CMP, IP (23.1%) or COP (13.0%) followed in order. The CMP group tended to be older, male, laborers, less educated, high-risk consumers of alcohol and smoking, and had a lower income status compared with the non-CMP group. We found that higher consumption of calcium, potassium, and fruits was negatively associated with CMP, while unhealthy dietary habits such as irregular meals and skipping breakfast were positively associated with CMP.

A previous representative study showed that the Korean population has a high incidence of multimorbidity pattern, including chronic disease such as cardiovascular disease [11,33]. Prevalence of multimorbidity was seen in 26.9% of Korean adults over the age of 40 years [33] and 86% of Korean adults over the age of 65 years [11]. Our results are unique in showing that multimorbidity patterns among Korean adults were CMP, IP, and COP, which is consistent with previous studies in other countries [3,34], although about 70% of adults have at least one multimorbidity pattern. A global population-based study presented that the most frequent patterns of multimorbidity across countries were cardiovascular and/or metabolic conditions [3,34]. In agreement with previous research studies [35,36], we found that CMP was the largest pattern found in Korean adults, which is linked to advanced age, and lower education and income than non-CMP individuals.

Our analysis suggest that diet is a crucial factor in multimorbidity. Higher consumption of fruits was associated with a lower prevalence of CMP. According to the Jiangsu Nutrition Study [16] of Chinese adults, the consumption of fruits, vegetables, and whole-grain products were associated with healthier stages among multimorbidity, such as coronary heart disease, stroke, hypertension and diabetes. The possible biological reason for this could be the phytochemicals and micronutrients present in fruits [16]. These compounds increase the antioxidant capacity of serum and increase the formation of endothelial prostacyclin that prevents platelet aggregation and reduces vascular tone [37]. Fruit and vegetable consumption is also associated with lower blood pressure and lower cholesterol and lipid level, which are the main risk factors for cardiovascular disease [38]. Thus, our results prove evidence-based beneficial effects on cardiometabolic health that suggest that dietary factors are associated with the presence of CMP. A total of 45.4% of deaths (a total of 702,308 cardiometabolic deaths in US adults) from heart disease, stroke, and type 2 diabetes were estimated to have a positive association with dietary factors, including low intake of fruit, vegetable, nut/seed and seafood, and high consumption of processed meat, sugar-sweetened beverages and sodium [18]. This is consistent with the fact that previous global analyses link cardiometabolic deaths with suboptimal dietary factors like excess sodium and SSBs, and low PUFA or consumption of fruits and vegetables. The CMP group consumed less sugar and sweetener (about 2 g on average), but more beverages (about 30 g on average) than the non-CMP group. A previous study showed that high levels of sugar-sweetened drinks or soft drink consumption is a risk factor of multimorbidity and increased multiple chronic diseases [39]. The mechanisms underlying the results linking CMP and beverage consumption could be related to the rapid absorption and metabolic reaction of simple sugars in beverages [40].

With respect to nutrients, we found that calcium and potassium were associated with lower CMP. Calcium and potassium are effective in blood pressure regulation, and prevent cardiovascular diseases and other health problems when combined with other essential nutrients (constituents of a varied diet) [41,42]. A higher consumption of calcium has been associated with beneficial effects on body weight [43], decreased waist circumference [44] and blood pressure [45]. Consistently, dietary intake of protein, calcium, potassium, and magnesium may mediate the blood-pressure-lowering effect [46]. Not only is there a clinically meaningful blood-pressure-lowering effect, it also could increase sodium excretion by the kidneys [47]. A high intake of potassium might influence the prevalence of CMP, which is known to be involved in sodium excretion. Likewise, dietary sodium was not significantly associated with CMP in our results, which is inconsistent with previous studies where high dietary sodium was related to CMP mortality [46,48]. The reason for the current result might be because the high-sodium-intake group ate more than twice the amount of vegetables, which has protective effects on CMP, compared to the low-sodium group (data not shown).

We showed that health-related behaviors including meal frequency, smoking, and alcohol consumption, are linked to CMP. Previous studies suggested that daily breakfast consumption as well as the number of meals may help to prevent cardiometabolic disease by decreasing the risk of adverse effects due to glucose and insulin metabolism [19,49]. Meanwhile, consumption of alcohol was a risk factor in men with prediabetes (OR = 1.49, 1.00–2.24). Low physical activity is also known to be a factor in the development of metabolic syndrome [13], which is not consistent with our results, Smoking [14] and high-risk alcohol consumption [15] are well-known risk factors for the development of metabolic disorders. Particularly, current smokers (20 cigarettes per day ≥) were observed to be more likely to have a prevalence of metabolic syndrome (OR = 2.24, 1.00–4.99) compared to people who never smoked [14]. A cohort study reported that unhealthy lifestyle factors increased the likelihood of multimorbidity in both men (5.23, 1.70–16.1) and women (1.95, 1.05–3.62) [15]. Our results thus suggest that irregular meals and skipping breakfast are associated with higher CMP, as well as smoking and high-risk alcohol drinking, which are risk factors for CMP.

This study has several strengths. We used data from a recent nationally representative dataset in South Korea and took into account its complex sampling design to provide representative estimates. It is also the first study to estimate multimorbidity patterns in South Korean adults aged from 19 to 64 years and to characterize multimorbidity in the non-elderly. Finally, our study considered various dietary factors as well as other lifestyle factors to facilitate multidimensional interpretation. Despite these strengths, this study has some limitations. First, our study is a cross-sectional design, and the results cannot suggest a cause–effect relationship. Second, this study could not include the duration of chronic diseases due to a lack of data. Third, dietary intake assessed by 24-h recall might not be representative of a typical subject’s intake. Further study will be needed to evaluate using standard methods to obtain the real intakes of the subjects.

## 5. Conclusions

In conclusion, we found that CMP accounts for more than half of the multimorbidity patterns in Korean adults aged 19 to 64 years and highlighted that CMP is associated with calcium, potassium, and fruit intake while unhealthy lifestyle habits such as irregular meals and skipping breakfast, as well as smoking and alcohol drinking, were found to be associated with higher CMP. These findings reinforce the concept that diet and healthy lifestyle factors are beneficial to health care and present evidence for multimorbidity prevention and management. Further investigation into the mechanisms underlying the role of these factors in the development of different multimorbidity patterns is needed.

## Figures and Tables

**Figure 1 nutrients-12-02730-f001:**
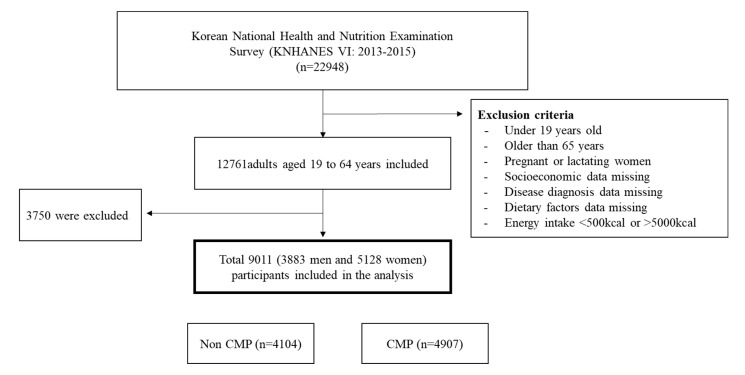
Flow chart for participant selection. KNHANES: Korean National Health and Nutrition Examination Survey; CMP: Cardiometabolic multimorbidity pattern.

**Figure 2 nutrients-12-02730-f002:**
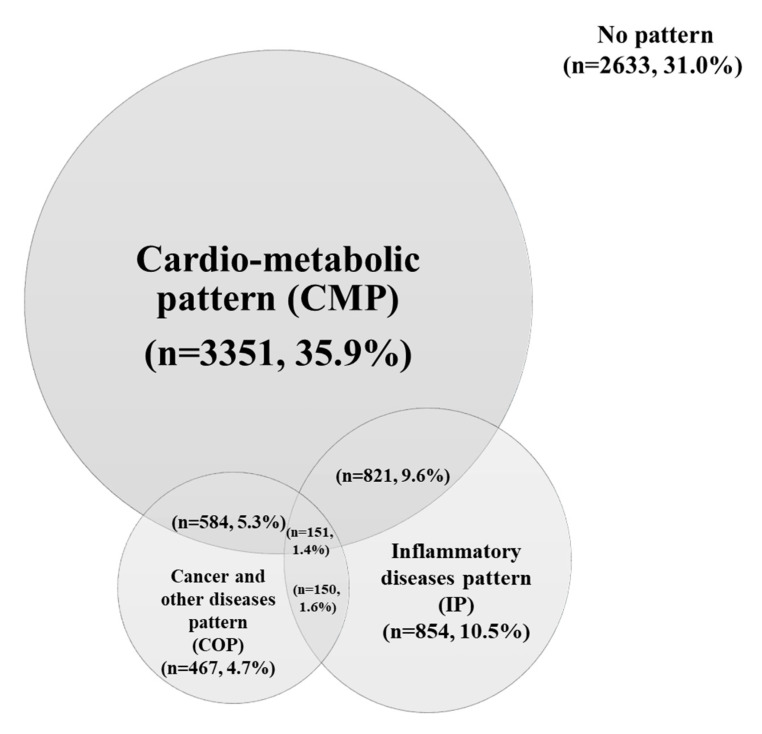
Overlapping multimorbidity patterns among adults aged 19–64 years in South Korea. All analysis accounting for the complex weights were used. Values are presented as *n*, %.

**Table 1 nutrients-12-02730-t001:** Sociodemographic characteristics and health-related behaviors of adults aged from 19 to 64 years in South Korea according to CMP (KNHANES VI 2013–2015) *.

Variables	Non-CMP	CMP	*p*-Value ^†^
	(*n* = 4104)	(*n* = 4907)	
Age	36.27 ± 0.21	45.07 ± 0.24	<0.0001
Sex (*n*,%)			
Male	1379 (41.65)	2504 (58.87)	<0.0001
Female	2725 (58.35)	2403 (41.13)	
Region (*n*,%)			
Urban	3046 (74.11)	3456 (71.09)	0.013
Rural	1058 (25.89)	1451 (28.91)	
Education (*n*, %)			
≤Elementary school	147 (2.92)	685 (10.99)	<0.0001
Middle school	215 (4.45)	626 (11.06)	
High school	1728 (44.56)	1876 (40.32)	
≥College	2014 (48.08)	1720 (37.63)	
Income (*n*,%) ^1^			
Lowest	890 (22.96)	1278 (26.06)	0.010
Low-middle	1013 (25.14)	1264 (26.10)	
Middle-high	1088 (25.77)	1192 (23.93)	
Highest	1113 (26.12)	1173 (23.91)	
Occupation (*n*,%) ^2^			
Office worker	2016 (48.68)	2014 (42.91)	<0.0001
Laborer	678 (17.00)	1426 (29.17)	
Unemployed	1410 (34.31)	1467 (27.92)	
Physical activity (*n*,%) ^3^			
Inactive	1518 (35.34)	1808 (34.98)	0.330
Active	1904 (46.53)	2219 (45.47)	
Health enhancing	682 (18.13)	880 (19.55)	
Smoking status (*n*, %)			
Current-smoker	696 (19.67)	1214 (28.74)	<0.0001
Ex-smoker	581 (15.70)	1039 (22.48)	
Non-smoker	2819 (64.63)	2627 (48.78)	
Alcohol intake (*n*,%) ^4^			
Low risk	3566 (86.59)	3979 (79.21)	<0.0001
High risk	538 (13.41)	928 (20.79)	

* All analyses accounting for complex weights were used to obtain nationally representative data. Values are presented as mean ± standard error or *n* (%). ^†^
*p*-values show differences between two groups (*p* < 0.05) Bold values denote a *p*-value < 0.05. ^1^ Income was divided based on quartile of individual income. ^2^ Occupations were integrated by shift pattern of duties (office worker: administration, profession, office worker, sales; service/laborer: agricultural, piscatorial, technology worker; laborer/unemployed: homemaker, student, unemployed). ^3^ Physical activity was divided into MET scores: Inactive < 600 MET score; 600 MET score ≤ active < 3000 MET score; health-enhancing ≥ 3000 MET score). ^4^ Risk was evaluated by calculating alcohol consumption per week (high-risk: 14 cups/wk > men; 7 cups/wk > women). CMP, Cardiometabolic Multimorbidity Pattern; KNHANES, Korea National Health and Nutrition Examination Survey.

**Table 2 nutrients-12-02730-t002:** Mean daily consumption of food and nutrients according to CMP among adults aged from 19 to 64 years in South Korea (KNHANES VI 2013–2015) *.

Variables	Non-CMP	CMP	*p*-Value ^†^
	(*n* = 4104)	(*n* = 4907)	
Nutrients			
Energy (kcal)	2117.10 ± 15.20	2118.08 ± 15.33	
Percentage from energy			
Carbohydrates (%)	63.8 ± 0.20	63.8 ± 0.21	
Protein (%)	14.7 ± 0.09	14.8 ± 0.08	
Fat (%)	21.5 ± 0.16	21.4 ± 0.16	
Carbohydrates (g)	318.22 ± 1.46	313.95 ± 1.57	0.050
Protein (g)	74.97 ± 0.49	74.71 ± 0.43	
Fat (g)	49.98 ± 0.44	49.83 ± 0.40	
Cholesterol (mg)	277.34 ± 4.23	277.85 ± 4.21	
Fiber (g)	21.99 ± 0.24	22.66 ± 0.27	
Calcium (mg)	515.83 ± 4.82	501.21 ± 4.69	0.030
Phosphorus (mg)	1134.60 ± 6.00	1123.77 ± 5.52	
Iron (mg)	17.89 ± 0.23	18.06 ± 0.38	
Sodium (mg)	4132.09 ± 42.11	4111.74 ± 39.02	
Potassium (mg)	3164.58 ± 23.58	3106.58 ± 22.06	
Vitamin A (μg RE)	768.6 ± 17.53	755.5 ± 16.86	
Thiamine (mg)	2.11 ± 0.01	2.1 ± 0.01	
Riboflavin (mg)	1.47 ± 0.01	1.44 ± 0.01	
Niacin (mg)	17.45 ± 0.13	17.31 ± 0.13	
Vitamin C (mg)	102.13 ± 2.25	99.6 ± 2.39	
Food group			
Cereals (g)	298.09 ± 2.39	297.99 ± 2.79	
Potato and starches (g)	42.76 ± 1.92	40.54 ± 2.00	
Sugar and sweeteners (g)	13.08 ± 0.44	11.51 ± 0.34	0.000
Pulses (g)	35.38 ± 1.40	36.91 ± 1.73	
Nuts and seeds (g)	8.8 ± 0.62	7.46 ± 0.58	
Vegetables (g)	326.88 ± 4.07	325.89 ± 4.02	
Fungi and mushrooms (g)	6.4 ± 0.37	6.89 ± 0.48	
Fruits (g)	208.7 ± 5.19	192.07 ± 5.40	0.020
Meats (g)	111.43 ± 2.45	112.41 ± 2.32	
Eggs (g)	29.58 ± 0.84	30.12 ± 0.85	
Fish and shellfish (g)	100.67 ± 3.49	97.23 ± 2.66	
Seaweeds (g)	25.92 ± 1.94	24.02 ± 1.59	
Milks (g)	90.2 ± 2.79	88.42 ± 2.96	
Oil and fat (g)	9.26 ± 0.19	9.63 ± 0.19	
Beverages ^§^ (g)	342.62 ± 8.08	370.13 ± 10.01	0.050
Seasonings (g)	41.17 ± 1.14	41.35 ± 1.02	
Processed foods (g)	0.23 ± 0.15	0.46 ± 0.18	

* All analyses accounting for complex weights were used to obtain nationally representative data. Values are presented as mean ± standard error. ^†^ Statistical analysis was performed using Student’s *t*-tests after adjustment for age, sex, and energy intake. Bold values denote a *p*-value < 0.05. CMP, Cardiometabolic Multimorbidity Pattern; KNHANES, Korea National Health and Nutrition Examination Survey. ^§^ Beverages group was included sugar-sweetened beverages (SSBs), non-sugar beverages, soft drinks, coffee, tea and alcoholic beverages based on the food groups categories of the KNHANES.

**Table 3 nutrients-12-02730-t003:** Associations (ORs and 95% CIs) between tertiles of nutrients and food consumption and CMP among adults aged from 19 to 64 years in South Korea (KNHANES 2013–2015) *.

Variables	Crude		Model 1 ^†^		Model 2 ^‡^	
	OR	95% CI	OR	95% CI	OR	95% CI
**Nutrients**						
**PUFAs** ^§^ **(g)**						
**Tertile 1 (Ref)**	**1**	**-**	**1**	**-**	**1**	**-**
Tertile 2	1.215	(1.076–1.373)	**0.827**	**(0.723–0.946)**	0.875	(0.763–1.003)
Tertile 3	1.217	(1.068–1.388)	**0.843**	**(0.714–0.995)**	0.912	(0.770–1.081)
*p* for trend	**0.003**		**0.02**		0.157	
**Dietary fiber (g)**						
Tertile 1 (Ref)	1	-	1	-	1	-
Tertile 2	**0.867**	**(0.767–0.980)**	**0.855**	**(0.746–0.979)**	0.901	(0.784–1.064)
Tertile 3	**0.652**	**(0.579–0.735)**	**0.823**	**(0.701–0.967)**	0.901	(0.763–1.064)
*p* for trend	**<0.0001**		**0.038**		0.316	
**Calcium (mg)**						
Tertile 1 (Ref)	1	-	1	-	1	-
Tertile 2	0.983	(0.870–1.111)	**0.781**	**(0.656–0.930)**	**0.843**	**(0.734–0.969)**
Tertile 3	0.903	(0.802–1.017)	**0.814**	**(0.676–0.980)**	**0.809**	**(0.691–0.945)**
*p* for trend	0.17		**0.015**		**0.018**	
**Sodium (mg)**						
Tertile 1 (Ref)	1	-	1	-	1	-
Tertile 2	1.019	(0.896–1.158)	**0.833**	**(0.724–0.959)**	**0.866**	**(0.751–0.997)**
Tertile 3	**0.870**	**(0.700–0.983)**	0.86	(0.730–1.013)	0.901	(0.763–1.063)
*p* for trend	**0.008**		**0.038**		0.135	
**Potassium (mg)**						
Tertile 1 (Ref)	1	-	1	-	1	-
Tertile 2	0.963	(0.849–1.092)	**0.765**	**(0.663–0.883)**	**0.804**	**(0.696–0.929)**
Tertile 3	**0.775**	**(0.685–0.876)**	**0.759**	**(0.639–0.901)**	**0.838**	**(0.704–0.998)**
*p* for trend	**<0.0001**		**0.001**		**0.013**	
**Foods**						
**Cereals (g)**						
Tertile 1 (Ref)	1	-	1	-	1	-
Tertile 2	0.926	(0.826–1.040)	0.901	(0.796–1.021)	0.921	(0.813–1.044)
Tertile 3	**0.823**	**(0.736–0.920)**	0.921	(0.793–1.069)	0.953	(0.818–1.110)
*p* for trend	**0.003**		0.255		0.385	
**Fruits (g)**						
Tertile 1 (Ref)	1	-	1	-	1	-
Tertile 2	**1.205**	**(1.069–1.358)**	**0.769**	**(0.675–0.877)**	**0.818**	**(0.717–0.933)**
Tertile 3	1.071	(0.952–1.205)	**0.759**	**(0.666–0.865)**	**0.841**	**(0.736–0.960)**
*p* for trend	0.009		**<0.0001**		**0.001**	
**Vegetables (g)**						
Tertile 1 (Ref)	1	-	1	-	1	-
Tertile 2	**0.857**	**(0.757–0.971)**	**0.836**	**(0.729–0.960)**	**0.857**	**(0.746–0.985)**
Tertile 3	**0.642**	**(0.566–0.728)**	0.877	(0.751–1.024)	**0.911**	**(0.778–1.067)**
*p* for trend	**<0.0001**		**0.04**		0.091	
**Meats (g)**						
Tertile 1 (Ref)	1	-	1	-	1	-
Tertile 2	**1.300**	**(1.149–1.470)**	**0.872**	**(0.760–1.000)**	0.912	(0.793–1.049)
Tertile 3	**1.423**	**(1.253–1.616)**	0.881	(0.761–1.021)	0.911	(0.784–1.059)
*p* for trend	**<0.0001**		0.121		0.376	

* All analyses accounting for complex weights were used to obtain nationally representative data. Bold indicates significance at a *p*-value < 0.05. ^†^ Model 1 adjusted for age, sex, and energy intake. **^‡^** Model 2 adjusted for age, sex, energy intake, income, education, physical activity, alcohol intake, and smoking status. CMP, Cardiometabolic Multimorbidity Pattern; KNHANES, Korea National Health and Nutrition Examination Survey; PUFAs, polyunsaturated fatty acids. ^§^ PUFA, *n*-3 fatty acid (*n*-3 FA), and *n*-6 fatty acid (*n*-6 FA) levels were included.

**Table 4 nutrients-12-02730-t004:** Associations (ORs and 95% CIs) between health-related behaviors including dietary habits and CMP among adults aged from 19 to 64 years in South Korea (KNHANES 2013–2015) *.

Variables	Crude		Model 1 ^†^		Model 2 ^††^	
	OR	95% CI	OR	95% CI	OR	95% CI
**Dietary habits**						
**Meal frequency**						
3 times a day (Ref)	1	-	1	-	1	-
2 times a day	**0.779**	**(0.700–0.867)**	**1.217**	**(1.083–1.367)**	**1.164**	**(1.034–1.312)**
Once a day	0.780	(0.558–1.091)	**1.509**	**(1.043–2.185)**	1.392	(0.950–2.041)
**Breakfast frequency**						
5–7 times a week (Ref)	1	-	1	-	1	-
3–4 times a week	**0.632**	**(0.542–0.737)**	1.027	(0.870–1.212)	1.020	(0.863–1.206)
1–2 times a week	**0.760**	**(0.652–0.885)**	**1.326**	**(1.118–1.573)**	**1.279**	**(1.078–1.518)**
Less than once a week	**0.697**	**(0.600–0.810)**	1.131	(0.960–1.332)	1.060	(0.898–1.251)
**Eat out frequency**						
More than once a day	**0.702**	**(0.603–0.871)**	**0.822**	**(0.689–0.980)**	0.895	(0.745–1.076)
1~6 times a week	**0.649**	**(0.571–0.739)**	**0.846**	**(0.735–0.973)**	0.910	(0.786–1.054)
Less than once a week (Ref)	1	-	1	-	1	-
**Health-related behaviors**						
**Physical activity**						
Inactive (Ref)	1	-	1	-	1	-
Active	0.987	(0.890–1.095)	1.020	(0.910–1.144)	1.054	(0.939–1.182)
Health enhancing	1.089	(0.944–1.257)	1.098	(0.937–1.286)	1.118	(0.955–1.310)
**Smoking status**						
Current-smoker	**1.936**	**(1.716–2.185)**	**1.353**	**(1.151–1.591)**	**1.303**	**(1.108–1.533)**
Ex-smoker	**1.897**	**(1.656–2.173)**	1.061	(0.892–1.263)	1.063	(0.893–1.266)
Non-smoker (Ref)	1	-	1	-	1	-
**Alcohol drinking**						
High risk	**1.694**	**(1.487–1.931)**	**1.515**	**(1.314–1.747)**	**1.490**	**(1.292–1.718)**
Low risk (Ref)	1	-	1	-	1	-

* All analyses accounting for complex weights were used to obtain nationally representative data. Bold indicates significance at a *p*-value < 0.05. ^†^ Model 1 adjusted for age, sex, and energy intake. ^††^ Model 2 adjusted for age, sex, energy intake, income, education, physical activity, alcohol intake, and smoking status. CMP, Cardiometabolic Multimorbidity Pattern; KNHANES, Korea National Health and Nutrition Examination Survey.

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
