# Peer review of "Association of Cardiometabolic Multimorbidity Pattern with Dietary Factors among Adults in South Korea"

_nutrients, 2020, doi:10.3390/nu12092730_

Round 1
Reviewer 1 Report
Remarks
Line 95ff: Define multimorbidity, how many diseases must at least occur together, how are diseases defined, are risk factors included?
Section 2.2 Define how nutrient intake (calcium, sodium, potassium and PUFAs) was estimated from the database and which foods are the major contributors.
Line 75: Which type of food recall, how was the recall validated? Was this a one-time 24 hour recall or several times?
Line 89: Be more specific, what volume has a cup in this study?
Line 107: Specify NHNANES
Line 112: With respect to statistics “tetrachoric correlation matrix”, “scree plot” and “Kaiser–Meyer–Olkin method” would profit from a short explanation. Clearly state why this statistic was used instead of more familiar ANOVA statistic.
Line 120: Not clear whether CMP characterizes individuals with all the listed diseases, one of them, some of them?
Line 122ff: Just describe how categories were built, results belong to the respective section
Line 172: How were nutrients such as minerals and fatty acids calculated from the 24 h recall?
Line 177: Define beverages, with or without alcohol, caffeine, energy, sugar etc.
Table 3: Is this Table referring to “…food and nutrient intake …”?
Table 3: Define “PUFAs” and specify the fatty acids included in the analysis
Table 4: Why is “p for trend” not indicated?
Line 251: I do not see a proof, there are some “associations,” correct and rephrase this section
Line 253: Without a definition of “beverage” this statement is speculation. If the beverage was a sugary drink (30 g sugary drink equals about 3 g sugar) than both groups had the same daily sugar consumption.
Line 259ff: This section needs work, why is calcium not discussed? What is the relationship between sodium and potassium? Why would high vegetable intake be associated with high sodium (I would expect the other way)? What is contained in the data not shown?
Suggested revisions
The overall objectives of the manuscript are clear, and the authors address them. The value of the manuscript derives from the specific nutrient association with a yet incompletely defined compound multimorbidity pattern for cardio metabolic health.
The authors should explain (methods and discussion) why the chosen statistical method adds value to this sub analyses of the KNHANES study rather than a standard ANOVA statistic.
The authors should better define disease indications, multimorbidity markers and nutritional intake parameters (examples provided under remarks).
The discussion needs revisions and work on the section of CMP association with mineral intake and on the section discussing the association of CMP with sugar, beverage and fruit intake.
Provide a more comprehensive discussion of the limitations of this study. What are the limitations of compiling several diseases into a multimorbidity score? What assumptions had to be made when assessing the nutritional intake? What are the statistical limitations when correcting for 8 parameters (Model 2)?
It will be necessary to have the English language reviewed for grammar and clarity.
Author Response
Reviewer 1
Comments and Suggestions for Authors
Remarks
Line 95ff: Define multimorbidity, how many diseases must at least occur together, how are diseases defined, are risk factors included?
Answer : To clarify, we added the 2.4. Definition of Multimorbidity in revised manuscript as below:
2.4. Definition of Multimorbidity
Multimorbidity was defined as suffering from two or more of the chronic medical diseases.[1] The medical conditions used to define multimorbidity in this study, 24 chronic diseases including hypertension, dyslipidemia, stroke, cardiovascular disease (e.g., myocardial infarction, angina pectoris), osteoarthritis and osteoporosis, cataract, depression, asthma, pulmonary tuberculosis, hepatitis B, cancer, thyroid disease, anemia, sinusitis, allergic rhinitis, atopic dermatitis, and tympanitis using a combination of self-reported physician diagnosis and/or current use of a disease treatment medication, or health examination-based diagnosis: ~~ Intrinsic cardiometabolic risk factors (total cholesterol, LDL-cholesterol, VLDL-cholesterol, insulin, HDL:LDL ratio, HDL:TAG) were not included.
Section 2.2 Define how nutrient intake (calcium, sodium, potassium and PUFAs) was estimated from the database and which foods are the major contributors.
Answer : As you suggested, we defined how nutrient and food intake were calculated in detail in 2.2
2.2. Assessment of dietary factors
The mean daily total energy, nutrients and food intake were assessed by a 1-day 24-hour recall method based on the Korean Foods and Nutrients Database of the Rural Development Administration. Food groups were classified into 18 groups (grains, potatoes, sugars, beans, nuts, vegetables, mushrooms, fruits, meats, eggs, fish, seaweeds, dairy products, oils, beverages, seasonings, processed foods etc.)
Line 75: Which type of food recall, how was the recall validated? Was this a one-time 24 hour recall or several times?
Answer : As we answered above, dietary intake was assessed using the single 24-hour diet recall method through home interviews by trained interviewers. Since 1998, the KNHANES has been collected nutrition data by a 1-day 24-hour recall method from 70,769 Korean population. It is widely well-known method using multiple (three-day or seven days) 24-hr recalls in nutritional studies, nevertheless, the single 24-hour diet recall method has been validated and developed by Korean Centers for Disease Control for 22 years.
Line 89: Be more specific, what volume has a cup in this study?
Answer : In KNHANES, 1 unit of alcohol was calculated for Soju(Korean distilled spirits, 48ml, 20%), beer (220ml, 4.5%) and makgeolli (Korean traditional rice-wine, 200ml, 6%)).
Line 107: Specify NHNANES
Answer : We revised typing mistake NHANES to KNHANES in the revised manuscript.
Line 112: With respect to statistics “tetrachoric correlation matrix”, “scree plot” and “Kaiser–Meyer–Olkin method” would profit from a short explanation. Clearly state why this statistic was used instead of more familiar ANOVA statistic.
Answer : According to your opinion, we probably analyze the data by using ANOVA if the subjects who were simply divided into diagnosis of disease what they have already. However, we identified multimorbidity patterns by using exploratory factor analysis most widely used analytical approach to account for binary morbidity data. A tetrachoric correlation matrix will lead to more valid results1 to assess the correlation structure between the variables—in this case, to assess the correlation with each disease. Particularly due to the dichotomous nature of the variables, a tetrachoric correlation matrix was generated.
1 Kubinger KD. On artificial results due to using factor analysis for dichotomous variables. Psychol Sci 2003;45:106–10.
Line 120: Not clear whether CMP characterizes individuals with all the listed diseases, one of them, some of them?
Answer : We attached supplementary Table 1 to clarify whether CMP characterizes. Participants were assigned to multimorbidity patterns if they had at least three diagnosis groups with a factor loading of 0.25 (indicting a strong association) on the corresponding pattern. For example, a person who were assigned to into Factor 1(called CMP), highly correlated with obesity, dyslipidemia, hypertension, diabetes mellitus, osteoarthritis and osteoporosis, depression, stroke, and CVD.
Line 122ff: Just describe how categories were built, results belong to the respective section
Answer : Exploratory factor analysis revealed three multimorbidity patterns in 9,011 Korean adults aged 19 to 64 years, the factor scores of which after oblique rotation are shown in Supplementary Table 1. Pattern 1, which included obesity, dyslipidemia, hypertension, diabetes mellitus, osteoarthritis and osteoporosis, depression, stroke, and CVD, showed the highest factor score of 0.80 with hypertension. This pattern was characterized by cardiometabolic disease and depression, in agreement with the result of many review papers. Thus, the present study labeled pattern 1 as CMP. Pattern 2 included allergic rhinitis, sinusitis, atopic dermatitis, asthma, and otitis; this pattern was called the IP, which is the main cause of inflammation. Pattern 3 included cancer, anemia, thyroid disease, hepatitis B, and cataract. Cancer showed the highest explanatory power, with a factor score of 0.57 in pattern 3. Therefore, pattern 3 was labeled as COP. (Supplementary Table 1)
Supplementary Table 1. Factor scores for multimorbidity patterns in adults aged 19 to 64 years in South Korea (KNHANES VI 2013–2015)
Disease |
Factor 1 Cardiometabolic diseases pattern |
Factor 2 Inflammatory disease pattern |
Factor 3 Cancer and other diseases pattern |
Obesity |
0.58 |
-0.02 |
-0.15 |
Dyslipidemia |
0.64 |
-0.01 |
0.02 |
Hypertension |
0.80 |
-0.14 |
0.08 |
Allergic rhinitis |
-0.11 |
0.63 |
-0.14 |
Diabetes mellitus |
0.64 |
-0.10 |
0.03 |
Anemia |
-0.04 |
-0.08 |
0.29 |
Osteoarthritis and osteoporosis |
0.45 |
0.16 |
0.40 |
Depression |
0.26 |
0.21 |
0.17 |
Atopic dermatitis |
-0.16 |
0.48 |
-0.38 |
Pulmonary tuberculosis |
0.16 |
-0.05 |
0.21 |
Thyroid disease |
-0.02 |
0.07 |
0.37 |
Asthma |
0.21 |
0.51 |
-0.08 |
Sinusitis |
-0.05 |
0.60 |
0.21 |
Cancer 1) |
0.01 |
-0.07 |
0.57 |
Otitis |
-0.18 |
0.50 |
0.09 |
Stroke and CVD |
0.58 |
0.00 |
0.10 |
Hepatitis B |
0.04 |
0.07 |
0.28 |
Cataract |
0.36 |
0.20 |
0.46 |
Statistical analysis was performed using exploratory factor analysis
1) Cancer summed gastric, liver, colorectal, breast, uterine/cervical, lung, thyroid, and others
Bold indicates significance at factor score > 0.25
KNHANES: Korea National Health and Nutrition Examination Survey, CVD: cardiovascular disease
Line 172: How were nutrients such as minerals and fatty acids calculated from the 24 h recall?
Answer : The mean daily total energy, nutrients and food intake were assessed by a 1-day 24-hour recall method based on the Korean Foods and Nutrients Database of the Rural Development Administration.1 Like as other nutrient intake, minerals and fatty acids also were determined from the sum of the nutrient content of all foods that an individual participant consumed during the day. The measured nutrients included carbohydrate, fat, protein, fiber, vitamin A, beta-carotene, retinol, thiamin, riboflavin, niacin, and vitamin C. The intake of each nutrient was expressed as a continuous variable and referred to the total intake of that nutrient over the previous 24 hours.
1. Rural Development Administration. National Academy of Agricultural Science. Standard Food Composition Table. 8th rev. ed. Seoul: Kyomoonsa; 2011.
Line 177: Define beverages, with or without alcohol, caffeine, energy, sugar etc.
Answer : Following as your comment, we added the definition of the beverage group in the footnote of Table 2.
- Beverages group was included sugar-sweetened beverages (SSBs), non-sugar beverages, soft drinks, coffee, tea, and alcoholic beverages based on the food groups categories of the KNHANES.
Table 3: Is this Table referring to “…food and nutrient intake …”?
Answer : We revised the title of Table 3 as you pointed out.
Table 3. Associations (ORs and 95% CIs) between tertiles of nutrients and foods consumption and CMP among adults aged 19 to 64 years in South Korea (KNHANES 2013-2015)*.
Table 3: Define “PUFAs” and specify the fatty acids included in the analysis
Answer : We defined “PUFAs” and specify fatty acids included in the analysis in the footnote of Table 3.
- PUFA, n-3 fatty acid (n-3 FA), and n-6 fatty acid (n-6 FA) levels were included.
Table 4: Why is “p for trend” not indicated?
Answer : Because health-related behaviors were categorized into each of frequencies of the variables, therefore we only presented to odds ratio with 95% CI in crude, Model 1 and 2 in Table 4.
Line 251: I do not see a proof, there are some “associations,” correct and rephrase this section
Answer : We revised the paragraph as you recommended, connected to “associations” with dietary factors and multimorbidity/ mortality.
Line 268- 295
Our analysis suggest that diet is a crucial factor in multimorbidity. Higher consumption of fruits was associated with lower prevalence of CMP. According to the Jiangsu Nutrition Study [16] of Chinese adults, consumption of fruits, vegetables, and whole grain products were associated with healthier stages among multimorbidity such as coronary heart disease, stroke, hypertension, and diabetes. The possible biological reason could be explained by the phytochemicals and micronutrients present in fruits [16]. These compounds increase the antioxidant capacity of serum and increase the formation of endothelial prostacyclin that prevents platelet aggregation and reduces vascular tone [37]. Fruit and vegetable consumption is also associated with lower blood pressure and lower cholesterol and lipid level, which are main risk factors for cardiovascular disease [38]. Thus, our results provide evidence-based beneficial effects on cardiometabolic health that suggests dietary factors are associated with the presence of CMP. Especially, 45.4% of deaths mortality (a total of 702,308 cardiometabolic deaths in US adults) from heart disease, stroke, and type 2 diabetes was estimated to have a positive association with dietary factors including low intakes of fruit, vegetable, nut/seed, seafood and high consumption of processed meat, sugar-sweetened beverages, and sodium [18]. It is consistent with that previous global analyses of the cardiometabolic deaths with suboptimal dietary factors like as excessed sodium and SSBs, low PUFA or fruits and vegetables. The CMP group consumed less sugar and sweetener (about 2 g on average), but higher beverage (about 30 g on average) than the non-CMP group. A previous study high levels of sugar-sweetened drinks or soft drink consumption is a risk factor of multimorbidity and increased the multiple chronic diseases [39]. The mechanisms underlying the results between CMP and beverage consumption could be related to the rapid absorption and metabolic reaction of simple sugars in beverages [40].
Line 253: Without a definition of “beverage” this statement is speculation. If the beverage was a sugary drink (30 g sugary drink equals about 3 g sugar) than both groups had the same daily sugar consumption.
Answer : As you suggested that of definition of “beverage”, we added a footnote in Table 2. Sugar and sweeteners were included sugar, honey, and added nutritive sweeteners such as high-fructose corn syrup following as the food groups of the Korean Nutrient Database [21]. With respect to beverages were included sugar-sweetened beverages (SSBs), non-sugar beverages, soft drinks, coffee, tea, and alcoholic beverages, too. Therefore, the sugar content is different between sugar and sweeteners and beverage based on the metabolic fates and effects of in the human body.
Line 259ff: This section needs work, why is calcium not discussed?
Answer : We added calcium and cardiometabolic health benefits in discussion.
Line 299- 303
Higher consumption of calcium has been associated with beneficial effects on body weight [43], decreased waist circumference [44] and blood pressure [45]. Consistently, dietary intake of protein, calcium, potassium, and magnesium may mediate blood pressure lowering effect [46]. Not only clinically meaningful blood pressure lowering effect, also could increase sodium excretion by the kidneys [47].
What is the relationship between sodium and potassium?
Answer : With respect to consumption of sodium (Non-CMP : 4,132.09 ± 42.11 vs. CMP : 4,111.74± 39.02) and potassium(Non-CMP : 3,164.58 ± 23.58 vs. CMP : 3,106.58 ± 22.06), no group differences were observed between Non-CMP and CMP in Table 2. However, negative associations were shown between tertiles of nutrients and foods consumption food and nutrients and CMP in both crude and Model 1 (Table 3).
Why would high vegetable intake be associated with high sodium (I would expect the other way)? What is contained in the data not shown?
Answer : We fully understand your opinion to our result related to the association high sodium and vegetable intake. These results were explained by unique cooking method like as pickling or pickled vegetables with salts or vinegar in Asian preserving culture. Higher sodium consumption were shown in vegetable and pickle (Non-CMP: 102.17 ±3.10 vs. CMP: 100.58 ±6.59, p<.0001) than meat (Non-CMP: 60.38 ±2.04 vs. CMP: 62.41 ±5.26, p<.0001) among Korean adults.
Submission Date 14 July 2020
Date of this review 30 Aug 2020

Reviewer 2 Report
The authors presented a quite large cross-sectional study with a focus on cardiometabolic multimorbidity. However, I feel it does not give novel evidence in the field.
Some questions/ comments to the authors:
- The authors used BMI cut off point for obesity as 25kg/m2. Many times WHO suggested for many Asian populations, trigger points for
public health action as 23 kg/m2 or higher (overweight), and 27·5 kg/m2 or higher (obesity). - It would be more interesting if you compare subgroups according to age. It is obvious that the younger population had a lower cardiometabolic multimorbidity.
- Concerning no differences in physical activity, and higher intake of sodium in non-CMP group puts the statistical model in question.
- Please revise references 20, 21, and 27
- English requires editing.
Author Response
1. The authors used BMI cut off point for obesity as 25kg/m2. Many times WHO suggested for many Asian populations, trigger points for public health action as 23 kg/m2 or higher (overweight), and 27·5 kg/m2 or higher (obesity).
Answer : We followed the definition of KNHANES, Korean Society for the Study of Obesity, and revised version of the clinical guidelines1 for prevention and treatment of obesity in 2018 to identify obesity -related diseases and health problems. The most recent guidelines are based partly on an analysis of data from the Korean National Health Insurance Service Health Checkup database collected from 2006 to 2015. The data analyzed included a total of 84,690,131 Korean adults.2
Classification Body mass index (kg/m2)
Underweight <18.5
Normal 18.5–22.9
Pre-obese 23–24.9
Obese class I 25–29.9
Obese class II 30–34.9
Obese class III ≥35
* Pre-obese may be defined as overweight or at-risk weight, and obese class III may be defined as extreme obesity.
1.Korean Society for the Study of Obesity. Guideline for the management of obesity 2018. Seoul: Korean Society for the Study of Obesity; 2018.
2.Seo MH, Kim YH, Han K, Jung JH, Park YG, Lee SS, et al. Prevalence of obesity and incidence of obesity-related comorbidities in Koreans based on National Health Insurance Service Health Checkup Data 2006–2015. J Obes Metab Syndr. 2018;27:46–52. doi: 10.7570/jomes.2018.27.1.46.
2.It would be more interesting if you compare subgroups according to age. It is obvious that the younger population had a lower cardiometabolic multimorbidity.
Answer : We agree to your opinion, however, in the present study we only focused to present national data and statistical power of the main outcomes related to association with CMP among Korean adults aged 19-64 years. Further study will be needed to compare subgroups according to age and age-standardized data.
3. Concerning no differences in physical activity, and higher intake of sodium in non-CMP group puts the statistical model in question.
Answer : As we mentioned above, we don’t include other risk factors (intrinsic or extrinsic) when we define multimorbidity to focus on diagnosed diseases only. Even we did not find any differences in the descriptive analysis (Table 1), there were possibilities exist interpersonal differences among individuals. Epidemiologic studies have found that a lack of adequate physical activity [13], smoking [14], high risk alcohol consumption [15], and unhealthy dietary intake [16] as potential risk factors to increase the risk of many chronic diseases. Accordingly, we consider individual differences (interpersonal) and their effects could be carried on the results with confounders- age, sex, energy intake, income, education, physical activity, alcohol intake, and smoking status.(Table 3 and 4).
4. Please revise references 20, 21, and 27
Answer : We revised references as below;
- Sciences, National Academy of Agricultural Sciences. Food Composition Table. 8th revision. 2011; p 1-636.
- Korean Nutrition Society. Dietary Reference Intakes for Koreans.
Seoul: Korean Nutrition Society; 2015
- World Health Organization, Regional Office for the Western Pacific, International Association for the Study of Obesity. International Obesity Task Force. The Asia-Pacific perspective: redefining obesity and its treatment. Melbourne. Health Communications Australia 2000.
5. English requires editing.
Answer : As your comment, we received English editing service for the revised manuscript.
Submission Date
14 July 2020
Date of this review
30 Aug 2020
